# MMVAE+: Enhancing the Generative Quality of Multimodal VAEs without Compromises

**Emanuele Palumbo, Imant Daunhawer & Julia E Vogt**
Department of Computer Science
ETH Zürich
Switzerland
`emanuele.palumbo@inf.ethz.ch`

## Abstract

Multimodal VAEs have recently gained attention as efficient models for weakly-supervised generative learning with a large number of modalities. However, all existing variants of multimodal VAEs are affected by a non-trivial trade-off between generative quality and generative coherence. We focus on the mixture-of-experts multimodal VAE (MMVAE), which achieves good coherence only at the expense of sample diversity and a resulting lack of generative quality. We present a novel variant of the MMVAE that improves its generative quality, while maintaining high semantic coherence. For this, shared and modality-specific information is modelled in separate latent subspaces. In contrast to previous approaches with separate subspaces, our model is robust to changes in latent dimensionality and regularization hyperparameters. We show that our model achieves both good generative coherence and high generative quality in challenging experiments, including more complex multimodal datasets than those used in previous works.

## 1 Introduction

Multimodal VAEs (Suzuki et al., 2016; Wu & Goodman, 2018; Shi et al., 2019; 2021; Sutter et al., 2020; 2021) are a promising class of models for weakly-supervised generative learning. These methodological advances enabled applications in multi-omics data integration (Lee & van der Schaar, 2021; Minoura et al., 2021) and tumor segmentation from multiple imaging modalities (Dorent et al., 2019). Several variants of scalable multimodal VAEs have been proposed (Wu & Goodman, 2018; Shi et al., 2019; 2021; Sutter et al., 2020; 2021; Lee & Pavlovic, 2021), whose performance is compared in terms of generative quality and generative coherence. While generative quality measures how well a model approximates the data distribution, generative coherence approximates the semantic coherence of generated samples across modalities. Ideally, an effective multimodal generative model should fulfill both of these performance aspects, but recent work (Daunhawer et al., 2022) shows that the predominant approaches exhibit a fundamental trade-off between the two criteria. This trade-off limits the utility of existing methods for more complex real-world settings.

In this work we propose the MMVAE+, a novel variant of the mixture-of-experts multimodal VAE (MMVAE, Shi et al., 2019). The proposed model significantly improves the diversity of the generated samples, enhancing the generative quality without sacrificing semantic coherence. To achieve this, we model shared and private—i.e. modality-specific—information in separate latent subspaces. We prove that the proposed model optimizes a valid multimodal ELBO (Lemma 1) and that—unlike for the MMVAE—learning shared information by optimising cross-modal reconstruction does not conflict with modelling private information (Lemma 2). Compared to the predominant models from the class of scalable multimodal VAEs, ours is the only model that achieves *both* convincing generative quality *and* generative coherence (Section 4.1.1). Compared to previous approaches with separate latent subspaces, our model is much more robust to changes in latent dimensionality and regularization hyperparameters (Section 4.1.2). Finally, we successfully apply our model to a challenging multimodal dataset of image and text pairs (Section 4.2), that was shown to be too complex for existing multimodal VAEs (Daunhawer et al., 2022).

## 2 RELATED WORK

Previous work has achieved outstanding results for multimodal generative learning targeting specialized tasks, such as image-to-image translation (Zhu et al., 2017; Choi et al., 2018) or text-to-image synthesis (Reed et al., 2016). Different from these specialized approaches, which are typically limited to a fixed number of modalities, multimodal VAEs have the advantage of being a general approach for generative learning with multiple heterogeneous data modalities. Scalable multimodal VAEs rely on the assumption that the joint encoder decomposes in terms of unimodal encoders and previous work proposed three different formulations for the joint encoder: the product-of-experts (MVAE, Wu & Goodman 2018), mixture-of-experts (MMVAE, Shi et al. 2019), and mixture-of-product-of-experts (MoPoE-VAE, Sutter et al. 2021). In a comparative analysis, Daunhawer et al. (2022) have demonstrated that none of these models fulfills both desired performance criteria, namely generative quality and generative coherence. While product-based models show low generative coherence, mixture-based models learn average predictions for private features, resulting in a lack of diversity, which negatively affects generative quality.

Subsequent work that builds on these approaches introduces additional regularization terms (Sutter et al., 2020; Hwang et al., 2021), hierachical latent spaces (M.Sutter & Vogt, 2021; Vasco et al., 2022; Wolff et al., 2022), or modality-specific latent subspaces (Sutter et al., 2020; Lee & Pavlovic, 2021). Yet, it is not clear whether these extensions overcome the fundamental tradeoff between generative quality and coherence. For instance, for models with modality-specific latent subspaces, we uncover an intricate dependency on hyperparameters, which raises fundamental questions with regards to model selection. We show that our proposed method—which also belongs to the class of methods with modality-specific latent subspaces—does *not* suffer from the same shortcomings.

## 3 METHODS

Given $M$ modalities $\boldsymbol{x}_{1:M} := \boldsymbol{x}_1, \ldots, \boldsymbol{x}_M$, we learn a generative model of the form $p_\Theta(\boldsymbol{x}_{1:M}, \boldsymbol{u}, \boldsymbol{w}) = p(\boldsymbol{u})p(\boldsymbol{w}) \prod_{m=1}^{M} p_{\theta_m}(\boldsymbol{x}_m|\boldsymbol{u}, \boldsymbol{w})$ where the latent variables $\boldsymbol{u}$ and $\boldsymbol{w}$ are independent and denote shared and private (i.e. modality-specific) information respectively, $p(\boldsymbol{u})$ and $p(\boldsymbol{w})$ are priors, and the likelihoods $p_{\theta_m}(\boldsymbol{x}_m|\boldsymbol{u}, \boldsymbol{w})$ are parameterized by neural networks. To enable the model to scale efficiently with the number of modalities, we make the assumption of a mixture-of-experts joint encoder $q_\Phi(\boldsymbol{u}, \boldsymbol{w}|\boldsymbol{x}_{1:M}) = \frac{1}{M} \sum_{m=1}^{M} (q_{\phi_m^u}(\boldsymbol{u}|\boldsymbol{x}_m)q_{\phi_m^w}(\boldsymbol{w}|\boldsymbol{x}_m))$ where in the following we will refer to $q_{\phi_m^u}(\boldsymbol{u}|\boldsymbol{x}_m)$ as the shared encoder for modality $m$, and to $q_{\phi_m^w}(\boldsymbol{w}|\boldsymbol{x}_m)$ as the private or modality-specific encoder for modality $m$.

Building on the MMVAE (see Appendix A), we propose the MMVAE+ objective:

$$\mathcal{L}_{\text{MMVAE+}}(\boldsymbol{x}_{1:M}) = \frac{1}{M} \sum_{m=1}^{M} \mathbb{E}_{\substack{q_{\phi_m^u}(\boldsymbol{u}|\boldsymbol{x}_m) \\ q_{\phi_m^w}(\boldsymbol{w}|\boldsymbol{x}_m)}} \left[ \log \left( \frac{p_{\theta_m}(\boldsymbol{x}_m, \boldsymbol{u}, \boldsymbol{w})}{q_\Phi(\boldsymbol{u}, \boldsymbol{w}|\boldsymbol{x}_{1:M})} \prod_{n \neq m} \mathbb{E}_{p_{\psi_n}(\tilde{\boldsymbol{w}}_n)} [p_{\theta_n}(\boldsymbol{x}_n|\boldsymbol{u}, \tilde{\boldsymbol{w}}_n)] \right) \right] \quad (1)$$

where $p_{\psi_1}(\tilde{\boldsymbol{w}}_1), \ldots, p_{\psi_M}(\tilde{\boldsymbol{w}}_M)$ are auxiliary priors specific to each modality, with parameterized variance. In Appendix A we prove the following two lemmas. The former states that the proposed model optimizes a valid multimodal ELBO and therefore belongs to the family of multimodal VAEs. The second one shows that, with the MMVAE+ objective and unlike for the MMVAE, learning shared information by optimising cross-modal reconstruction does not conflict with modelling private information. In addition, it explains why shared encoders learn information that is shared between modalities, while private encoders learn modality-specific features for each modality.

**Lemma 1.** *The MMVAE+ objective (Equation 1) is a valid lower bound on* $\log p_\Theta(\boldsymbol{x}_{1:M})$.

**Lemma 2.** *Computed gradients for the MMVAE+ objective with respect to private encoder parameters* $\phi_1^w, \ldots, \phi_M^w$ *do not depend on cross-modal reconstruction likelihood terms, while gradients with respect to shared encoder parameters* $\phi_1^u, \ldots, \phi_M^u$ *depend on the reconstruction likelihood of all modalities.*

As in Shi et al. (2019), a multi-sample version of the objective can provide a tighter bound on the log-evidence. In Appendix B we discuss the details of the tighter version of the objective, along with the possibility to include a $\beta$ hyperparameter (Higgins et al., 2017) to weight the KL-divergence term. Finally, note that the auxiliary priors $p_{\psi_1}(\tilde{\boldsymbol{w}}_1), \ldots, p_{\psi_M}(\tilde{\boldsymbol{w}}_M)$ are only used to compute meaningful

cross-modal reconstruction likelihoods at training time, but not for conditional generation at test time, as explained in Appendices A and G.

## 4 EXPERIMENTS

We report experimental results for two challenging datasets introduced in previous work, namely PolyMNIST (Sutter et al., 2021) and Caltech Birds (CUB) Image-Captions (Shi et al., 2019; Wah et al., 2011). While previous work (Shi et al., 2019; 2021) has tackled a simplified version of the latter experiment by using pretrained ResNet-features, we train the models on actual images—a more realistic setting that is too challenging for the predominant approaches (Daunhawer et al., 2022).

### 4.1 POLYMNIST

#### 4.1.1 ENHANCED GENERATIVE QUALITY FOR SEMANTICALLY COHERENT GENERATION

Figure 1 shows the results on the PolyMNIST dataset, comparing MMVAE+ to the predominant approaches, namely MVAE, MMVAE and MoPoE-VAE. Qualitative results for conditional generation (left subplot) highlight the shortcomings of existing approaches: while the MVAE shows almost no generative coherence, MMVAE and MoPoE-VAE both show a lack of diversity in modality-specific features, such as the background details or the style of the handwritten digit. In contrast, MMVAE+ shows convincing generative quality, with clear variation in private features for generated samples, while at the same time high semantic coherence. For a quantitative comparison, we compare generative coherence (Shi et al., 2019) and FID scores (Heusel et al., 2017) in a scatterplot, showing that only the MMVAE+ reaches high generative quality *and* high semantic coherence for both conditional and unconditional generation. Notably, the advantage of the MMVAE+ persists over a large range of hyperparameter values.

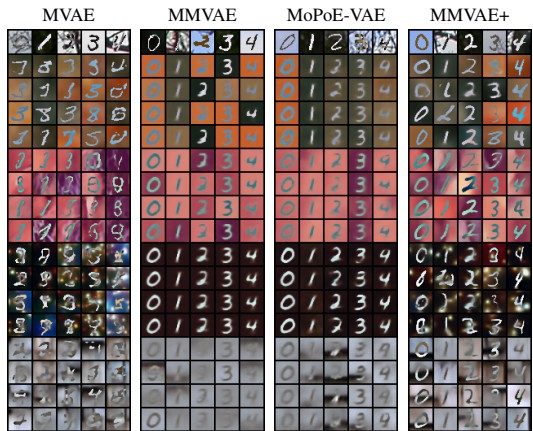 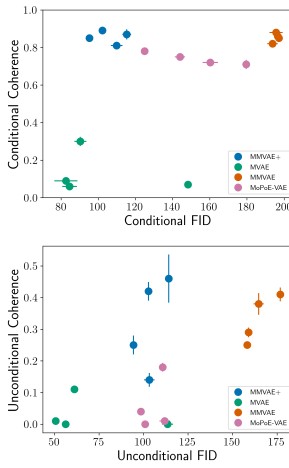

Figure 1: *Left:* qualitative results for conditional generation on the PolyMNIST dataset. Input sample from the first modality is shown in the top row, and below are four conditionally generated samples for each of the remaining modalities. We use $\beta = 1$ for all trained models. *Right:* comparison of generative quality (in terms of FID, lower is better) and generative coherence (higher is better). The upper scatterplot evaluates conditional generation and the lower quantifies unconditional generation performance. For each model, we show the results for a range of representative $\beta$ values (see Appendix D). An optimal model would lie in the top-left area of each scatterplot.

#### 4.1.2 COMPARING WITH ALTERNATIVE MODELS WITH SEPARATE LATENT SUBSPACES

In this section, we compare MMVAE+ with alternative variants of multimodal VAEs that use separate shared and modality-specific latent subspaces. Crucially, we find that the existing VAE-based

alternatives (Sutter et al., 2020; Lee & Pavlovic, 2021) exhibit a fundamental problem: to reach generative coherence, they require strong regularization of the modality-specific subspaces. In fact, these models require careful hyperparameter tuning to prevent a trivial *shortcut* where all information flows through the modality-specific subspaces and the shared subspace is ignored by the decoder. Figure 2 illustrates this problem with an ablation for MMJSD (Sutter et al., 2020). We discuss this problem in further detail in Appendix F. In contrast, the MMVAE+ does *not* suffer from the same shortcut problem and works without regularising modality-specific subspaces.

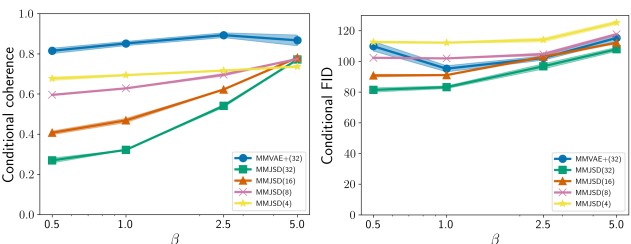 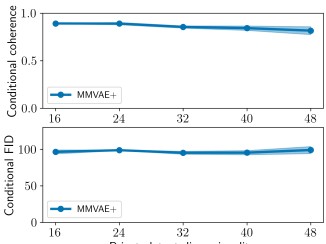

Figure 2: Illustration of the shortcut problem for the MMJSD model. For MMJSD, regularization can be controlled by varying the number of dimensions allocated to the modality-specific subspaces, or by tuning $\beta$. We show conditional coherence (left subplot) and FID score (right subplot) on PolyMNIST, for different $\beta$ values and different number of dimensions for modality-specific subspaces (in parentheses in the figure legend). The number of dimensions allocated to the shared subspace is fixed to 32. MM-VAE+ is kept as reference.

Figure 3: Ablation showing that the MMVAE+ does *not* suffer from the shortcut problem. The MM-VAE+ exhibits stable conditional coherence (top) and FID (bottom) despite varying private latent dimensionality. Shared latent dimensionality varies to have a fixed total of 64 latent dimensions, and $\beta$ is equal to 1.

## 4.2 CUB IMAGE-CAPTIONS

In this section we test the MMVAE+ on the CUB Image-Captions experiment (Shi et al., 2019; Netzer et al., 2011) and compare its performance with alternative models. Conditional image generation results in Figure 4 clearly show that none of the existing methods is able to achieve sufficient performance results, as highlighted in recent work (Daunhawer et al., 2022). On the other hand MMVAE+ achieves convincing results both in generative quality and generative coherence, despite the challenging nature of the experiment, where there is a sizeable amount of private information to be modelled, and a profound difference between the two data types.

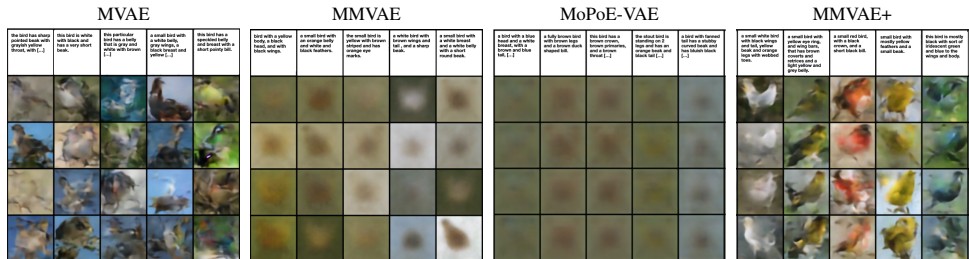

Figure 4: Conditional caption-to-image generation on the CUB Image-Caption dataset.

## 5 CONCLUSION

In this work we propose a new model in the family of multimodal VAEs, that considerably improves generative quality without compromising semantic coherence. We validate our approach on more complex datasets than the ones used to benchmark previous work, taking an important step towards the applicability of multimodal VAEs to real-world problems.

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

## A  PROOFS

The following two lemmas formalize two key properties of the MMVAE+. In the former, we prove that the MMVAE+ objective is a valid lower bound on the log-evidence, and therefore the proposed model belongs to the family of multimodal VAEs. In the latter, we demonstrate that learning shared information by maximising cross-modal reconstruction with the MMVAE+ objective does not conflict with modelling private information. In particular we elucidate how the desired information structure for learnt representations, where shared and private information is encoded in separate latent subspaces, is achieved.

**Lemma 1.** *Given the assumption of*

1. *a generative model for $M$ modalities $\boldsymbol{x}_{1:M} := \boldsymbol{x}_1, \ldots, \boldsymbol{x}_M$ of the form $p_\Theta(\boldsymbol{x}_{1:M}, \boldsymbol{u}, \boldsymbol{w}) = p(\boldsymbol{u})p(\boldsymbol{w}) \prod_{m=1}^{M} p_{\theta_m}(\boldsymbol{x}_m | \boldsymbol{u}, \boldsymbol{w})$ where the latent variables $\boldsymbol{u}$ and $\boldsymbol{w}$ are independent and denote shared and modality-specific information respectively, $p(\boldsymbol{u})$ and $p(\boldsymbol{w})$ are priors, and the likelihoods $p_{\theta_m}(\boldsymbol{x}_m | \boldsymbol{u}, \boldsymbol{w})$ are parameterized by neural networks*

2. *a mixture-of-experts joint encoder $q_\Phi(\boldsymbol{u}, \boldsymbol{w} | \boldsymbol{x}_{1:M}) = \frac{1}{M} \sum_{m=1}^{M} (q_{\phi_m^u}(\boldsymbol{u} | \boldsymbol{x}_m) q_{\phi_m^w}(\boldsymbol{w} | \boldsymbol{x}_m))$ where $q_{\phi_m^u}(\boldsymbol{u} | \boldsymbol{x}_m)$ is the shared encoder for modality $m$ and $q_{\phi_m^w}(\boldsymbol{w} | \boldsymbol{x}_m)$ is the private (or modality-specific) encoder for modality $m$*

*the MMVAE+ objective*

$$\mathcal{L}_{\text{MMVAE+}}(\boldsymbol{x}_{1:M}) = \frac{1}{M} \sum_{m=1}^{M} \mathbb{E}_{\substack{q_{\phi_m^u}(\boldsymbol{u}|\boldsymbol{x}_m) \\ q_{\phi_m^w}(\boldsymbol{w}|\boldsymbol{x}_m)}} \left[ \log \left( \frac{p_{\theta_m}(\boldsymbol{x}_m, \boldsymbol{u}, \boldsymbol{w})}{q_\Phi(\boldsymbol{u}, \boldsymbol{w}|\boldsymbol{x}_{1:M})} \prod_{n \neq m} \mathbb{E}_{p_{\psi_n}(\tilde{\boldsymbol{w}}_n)}[p_{\theta_n}(\boldsymbol{x}_n | \boldsymbol{u}, \tilde{\boldsymbol{w}}_n)] \right) \right]$$

(2)

*is a valid lower bound on $\log p_\Theta(\boldsymbol{x}_{1:M})$.*

*Proof.* Given our generative model, computing the log-evidence $\log p_\Theta(\boldsymbol{x}_{1:M})$ is intractable as one would have to have knowledge of the true posterior $p_\Theta(\boldsymbol{u}, \boldsymbol{w} | \boldsymbol{x}_{1:M})$. Therefore, we approximate the true posterior with a variational encoder $q_\Phi(\boldsymbol{u}, \boldsymbol{w} | \boldsymbol{x}_{1:M})$ obtaining the well-known VAE ELBO (Kingma & Welling, 2014)

$$\mathcal{L}_{\text{VAE}}(\boldsymbol{x}_{1:M}) = \mathbb{E}_{q_\Phi(\boldsymbol{u}, \boldsymbol{w}|\boldsymbol{x}_{1:M})} \left[ \log p_\Theta(\boldsymbol{x}_{1:M} | \boldsymbol{u}, \boldsymbol{w}) \right] - D_{KL}(q_\Phi(\boldsymbol{u}, \boldsymbol{w}|\boldsymbol{x}_{1:M}) \| p(\boldsymbol{u}, \boldsymbol{w})) \quad (3)$$

$$= \mathbb{E}_{q_\Phi(\boldsymbol{u}, \boldsymbol{w}|\boldsymbol{x}_{1:M})} \left[ \log \frac{p_\Theta(\boldsymbol{x}_{1:M}, \boldsymbol{u}, \boldsymbol{w})}{q_\Phi(\boldsymbol{u}, \boldsymbol{w}|\boldsymbol{x}_{1:M})} \right] \quad (4)$$

$$\leq \log p_\Theta(\boldsymbol{x}_{1:M})$$

The assumption of a mixture-of-experts joint encoder allows to write the expectation in (4) as a sum of expectations indexed by the $M$ unimodal encoders, obtaining the objective

$$\mathcal{L}_{\text{MoE}}(\boldsymbol{x}_{1:M}) = \frac{1}{M} \sum_{m=1}^{M} \mathbb{E}_{\substack{q_{\phi_m^u}(\boldsymbol{u}|\boldsymbol{x}_m) \\ q_{\phi_m^w}(\boldsymbol{w}|\boldsymbol{x}_m)}} \left[ \log \frac{p_\Theta(\boldsymbol{x}_{1:M}, \boldsymbol{u}, \boldsymbol{w})}{q_\Phi(\boldsymbol{u}, \boldsymbol{w}|\boldsymbol{x}_{1:M})} \right] \quad (5)$$

Note this is essentially the MMVAE objective (Shi et al., 2019)

$$\mathcal{L}_{\text{MMVAE}}(\boldsymbol{x}_{1:M}) = \frac{1}{M} \sum_{m=1}^{M} \mathbb{E}_{q_{\phi_m}(\boldsymbol{z}|\boldsymbol{x}_m)} \left[ \log \frac{p_\Theta(\boldsymbol{x}_{1:M}, \boldsymbol{z})}{q_\Phi(\boldsymbol{z}|\boldsymbol{x}_{1:M})} \right]$$

with the small difference of having latent representations that are the combination of two latent subspaces. Note each term in the sum in (5) expands as

$$\mathbb{E}_{\substack{q_{\phi_m^u}(\boldsymbol{u}|\boldsymbol{x}_m) \\ q_{\phi_m^w}(\boldsymbol{w}|\boldsymbol{x}_m)}} \left[ \log \left( \frac{p(\boldsymbol{u}, \boldsymbol{w})}{q_\Phi(\boldsymbol{u}, \boldsymbol{w}|\boldsymbol{x}_{1:M})} p_{\theta_m}(\boldsymbol{x}_m | \boldsymbol{u}, \boldsymbol{w}) \prod_{n \neq m} p_{\theta_n}(\boldsymbol{x}_n | \boldsymbol{u}, \boldsymbol{w}) \right) \right]$$

With our assumptions on the generative model we can write

$$p_{\theta_n}(\boldsymbol{x}_n | \boldsymbol{u}, \boldsymbol{w}) = p_{\theta_n}(\boldsymbol{x}_n | \boldsymbol{u}) = \mathbb{E}_{p_{\psi_n}(\tilde{\boldsymbol{w}}_n)}[p_{\theta_n}(\boldsymbol{x}_n | \boldsymbol{u}, \tilde{\boldsymbol{w}}_n)] \quad (6)$$

where $p_{\psi_n}(\tilde{\boldsymbol{w}}_n)$ is a prior specific to modality $n$, with parameterized variance.[1] Note that the first equality follows from our assumptions, in that the information on a different target modality $\boldsymbol{x}_n$ that is present in $\boldsymbol{x}_m$ is limited to shared information between modalities, which is encoded in $\boldsymbol{u}$. The second equality instead follows from simple probability. Note that in practice the expectation in (6) is approximated with a 1-sample Monte Carlo estimator. From the fact that (5) is a valid ELBO, and the equality in (6), follows that the MMVAE+ objective is a lower bound on $\log p_{\Theta}(\boldsymbol{x}_{1:M})$, and therefore the model maximizes a valid ELBO. □

**Lemma 2.** *Computed gradients for the MMVAE+ objective with respect to private encoder parameters $\phi_1^w, \ldots, \phi_M^w$ do not depend on cross-modal reconstruction likelihood terms, while gradients with respect to shared encoder parameters $\phi_1^u, \ldots, \phi_M^u$ depend on the reconstruction likelihood of all modalities.*

*Proof.* Computing the gradients for the MMVAE+ objective with respect the private encoder network parameters $\phi_m^w$ for a given modality $m \in \{1, \ldots, M\}$ gives

$$
\nabla_{\phi_m^w} \mathcal{L}_{\text{MMVAE+}}(\boldsymbol{x}_{1:M}) = \nabla_{\phi_m^w} \Bigg( \mathbb{E}_{\substack{q_{\phi_m^u}(\boldsymbol{u}|\boldsymbol{x}_m) \\ q_{\phi_m^w}(\boldsymbol{w}|\boldsymbol{x}_m)}} \big[ \log\big( p_{\theta_m}(\boldsymbol{x}_m|\boldsymbol{u},\boldsymbol{w})p(\boldsymbol{u},\boldsymbol{w})\big)\big] +
$$

$$
\frac{1}{M}\sum_{r=1}^M \mathbb{E}_{\substack{q_{\phi_r^u}(\boldsymbol{u}|\boldsymbol{x}_r) \\ q_{\phi_r^w}(\boldsymbol{w}|\boldsymbol{x}_r)}} \Big[ \log\Big( \frac{1}{q_\Phi(\boldsymbol{u},\boldsymbol{w}|\boldsymbol{x}_{1:M})}\Big)\Big]\Bigg)
$$

$$
= \nabla_{\phi_m^w} F_{\theta_m,\Phi}(\boldsymbol{x}_{1:M})
$$

Note that the resulting function does not depend on decoder parameters $\{\theta_n\}_{n\neq m}$, i.e. does not depend on cross-modal reconstruction likelihoods. Computing the gradients with respect the shared encoder network parameters $\phi_m^u$ for a given modality $m \in \{1, \ldots, M\}$ instead gives [2]

$$
\nabla_{\phi_m^u} \mathcal{L}_{\text{MMVAE+}}(\boldsymbol{x}_{1:M}) = \nabla_{\phi_m^u} \Bigg( \mathbb{E}_{\substack{q_{\phi_m^u}(\boldsymbol{u}|\boldsymbol{x}_m) \\ q_{\phi_m^w}(\boldsymbol{w}|\boldsymbol{x}_m)}} \Big[ \log\Big( p_{\theta_m}(\boldsymbol{x}_m|\boldsymbol{u},\boldsymbol{w})p(\boldsymbol{u},\boldsymbol{w})
$$

$$
\prod_{n\neq m} \mathbb{E}_{p_{\psi_n}(\tilde{\boldsymbol{w}}_n)}[p_{\theta_n}(\boldsymbol{x}_n|\boldsymbol{u},\tilde{\boldsymbol{w}}_n)]\Big)\Big]
$$

$$
+ \frac{1}{M}\sum_{r=1}^M \mathbb{E}_{\substack{q_{\phi_r^u}(\boldsymbol{u}|\boldsymbol{x}_r) \\ q_{\phi_r^w}(\boldsymbol{w}|\boldsymbol{x}_r)}} \Big[ \frac{1}{q_\Phi(\boldsymbol{u},\boldsymbol{w}|\boldsymbol{x}_{1:M})}\Big]\Bigg)
$$

$$
= \nabla_{\phi_m^u} G_{\Theta,\Phi,\Psi}(\boldsymbol{x}_{1:M})
$$

which is a function of the reconstruction likelihood of all modalities given inference from modality $m$, through decoders parameterized by $\Theta = \{\theta_1, \ldots, \theta_M\}$. □

Lemma 2 explains why factorization in the latent space between shared information in $\boldsymbol{u}$ and private information in $\boldsymbol{w}$ is achieved. Shared encoders learn information from a given modality, based on useful it is to reconstruct all modalities. The information that can be inferred from a given modality and be useful to reconstruct all modalities is by definition the shared information across modalities. As a result, shared information is modelled in $\boldsymbol{u}$. Private encoders on the other hand learn information for a given modality given how useful it is to reconstruct *only* the same given modality. Therefore $\boldsymbol{w}$ would be prompt to contain both shared and private information. However, if the decoders retrieved some of the shared information from $\boldsymbol{w}$, cross-modal reconstruction likelihoods would be bad, as $\boldsymbol{w}$ is resampled from auxiliary priors for cross-modal reconstruction at training. As a result, the information modelled in $\boldsymbol{w}$ consists of modality-specific features. In conclusion, as confirmed by the empirical results, we achieve factorization between shared information encoded in $\boldsymbol{u}$ and private information encoded in $\boldsymbol{w}$ as a result of the optimized objective, rather than as a result of constraints on latent capacity or strong regularization (see Section 4.1.2 and Appendix F).

In addition, comparing the MMVAE+ objective to the MMVAE objective one can understand where improved modelling of private features originates. With the MMVAE objective the reconstruction

---

[1] We discuss the effect of this parametrization in Appendix G

[2] Note $\Psi = \{\psi_1, \ldots \psi_M\}$ here, and in the following.

likelihood of all modalities is maximized given a single latent code $\boldsymbol{z}$ inferred from a given modality $m$. Given the same $\boldsymbol{z}$ has to be used to reconstruct *all* modalities, it is prompt to contain only information that can be useful to reconstruct all modalities, i.e. shared information, whereas private information is disregarded. For MMVAE+ instead, we resample private information from auxiliary priors at training time for cross-modal reconstruction, in order to learn shared information across modalities, without conflicitng with modelling modality-specific variation.

Finally, this also explains why we resample private information from $\boldsymbol{w} \sim p(\boldsymbol{w})$ at test time for conditional generation. Auxiliary priors are only used at training time to have learning of shared information in $\boldsymbol{u}$ not conflicting with modelling private variation in $\boldsymbol{w}$. As a result, shared information is encoded $\boldsymbol{u}$, while high-quality private information is encoded in $\boldsymbol{w}$. Therefore, to have high-quality private information in the latent code, that results in high sample diversity and generative quality, we resample $\boldsymbol{w} \sim p(\boldsymbol{w})$ for conditional generation at test time (see also Appendix G).

## B  MULTI-SAMPLE ESTIMATOR AND $\beta$ HYPERPARAMETER IN THE OBJECTIVE

### B.1  MULTI-SAMPLE ESTIMATORS

As Shi et al. (2019) suggest for the MMVAE objective, a tighter bound on the log-evidence can be enforced if one uses a multi-sample estimator in the MMVAE+ objective. In fact the $K$-sample version of the objective in (1)

$$\mathcal{L}_{\text{MMVAE+}}^{ms}(\boldsymbol{x}_{1:M}) = \frac{1}{M}\sum_{m=1}^{M}\mathbb{E}_{\substack{\boldsymbol{u}^{1:K}\sim q_{\phi_m^u}(\boldsymbol{u}|\boldsymbol{x}_m)\\\boldsymbol{w}^{1:K}\sim q_{\phi_m^w}(\boldsymbol{w}|\boldsymbol{x}_m)}}\left[\log\frac{1}{K}\sum_{k=1}^{K}C_{\Phi,\Theta,\Psi}(\boldsymbol{x}_{1:M},\boldsymbol{u}^k,\boldsymbol{w}^k)\right]$$

with

$$C_{\Phi,\Theta,\Psi}(\boldsymbol{x}_{1:M},\boldsymbol{u}^k,\boldsymbol{w}^k) = \frac{p_{\theta_m}(\boldsymbol{x}_m,\boldsymbol{u}^k,\boldsymbol{w}^k)}{q_\Phi(\boldsymbol{u}^k,\boldsymbol{w}^k|\boldsymbol{x}_{1:M})}\prod_{n\neq m}\mathbb{E}_{p_{\psi_n}(\tilde{\boldsymbol{w}}_n)}[p_{\theta_n}(\boldsymbol{x}_n|\boldsymbol{u}^k,\tilde{\boldsymbol{w}}_n)]$$

is a tighter ELBO for $K > 1$, as it can be shown with Jensen's inequality. However, as multi-sample estimators can lead to undesirably high variance, Tucker et al. (2019) have proposed a doubly-reparameterized gradient estimator (DReG) to reduce the variance in estimated gradients for latent-variable models with multi-sample objectives. As done by Shi et al. (2019) for the experiments with the multi-sample version of the MMVAE objective, here we also employ the DReG estimator when training the MMVAE+ with the multi-sample version of its objective.

### B.2  BALANCING RECONSTRUCTION AND REGULARIZATION

In practice the two contributions in (3) are often balanced by weighting the KL-divergence term with a hyperparameter $\beta$

$$\mathcal{L}_{\text{VAE}_\beta}(\boldsymbol{x}_{1:M}) = \mathbb{E}_{q_\Phi(\boldsymbol{u},\boldsymbol{w}|\boldsymbol{x}_{1:M})}\big[\log p_\Theta(\boldsymbol{x}_{1:M}|\boldsymbol{u},\boldsymbol{w})\big] - \beta D_{KL}(q_\Phi(\boldsymbol{u},\boldsymbol{w}|\boldsymbol{x}_{1:M}) \,\|\, p(\boldsymbol{u},\boldsymbol{w}))$$

$$= \mathbb{E}_{q_\Phi(\boldsymbol{u},\boldsymbol{w}|\boldsymbol{x}_{1:M})}\left[\log\left(p_\Theta(\boldsymbol{x}_{1:M}|\boldsymbol{u},\boldsymbol{w})\left(\frac{p(\boldsymbol{u},\boldsymbol{w})}{q_\Phi(\boldsymbol{u},\boldsymbol{w}|\boldsymbol{x}_{1:M})}\right)^\beta\right)\right] \qquad (7)$$

From (7), with analogous assumptions as in Lemma 1, we can derive the $\beta$-weighted MMVAE+ objective, and its $K$-sample version

$$\mathcal{L}_{\text{MMVAE+}_\beta}^{ms}(\boldsymbol{x}_{1:M}) = \frac{1}{M}\sum_{m=1}^{M}\mathbb{E}_{\substack{\boldsymbol{u}^{1:K}\sim q_{\phi_m^u}(\boldsymbol{u}|\boldsymbol{x}_m)\\\boldsymbol{w}^{1:K}\sim q_{\phi_m^w}(\boldsymbol{w}|\boldsymbol{x}_m)}}\left[\log\frac{1}{K}\sum_{k=1}^{K}D_{\Phi,\Theta,\Psi}^\beta(\boldsymbol{x}_{1:M},\boldsymbol{u}^k,\boldsymbol{w}^k)\right]$$

where

$$D_{\Phi,\Theta,\Psi}^\beta(\boldsymbol{x}_{1:M},\boldsymbol{u}^k,\boldsymbol{w}^k) = \frac{p_{\theta_m}(\boldsymbol{x}_m|\boldsymbol{u}^k,\boldsymbol{w}^k)\big(p(\boldsymbol{u}^k,\boldsymbol{w}^k)\big)^\beta}{\big(q_\Phi(\boldsymbol{u}^k,\boldsymbol{w}^k|\boldsymbol{x}_{1:M})\big)^\beta}\prod_{n\neq m}\mathbb{E}_{p_{\psi_n}(\tilde{\boldsymbol{w}}_n)}[p_{\theta_n}(\boldsymbol{x}_n|\boldsymbol{u}^k,\tilde{\boldsymbol{w}}_n)]$$

which is the form of the objective we use to tune regularization in the latent space for MMVAE+ in our ablations. In the same way, for our $\beta$-ablations with the MMVAE we use the objective

$$\mathcal{L}^{ms}_{\text{MMVAE}_\beta}(\boldsymbol{x}_{1:M}) = \frac{1}{M} \sum_{m=1}^{M} \mathbb{E}_{\boldsymbol{z}^{1:K} \sim q_{\phi_m}(\boldsymbol{z}|\boldsymbol{x}_m)} \left[ \log \frac{1}{K} \sum_{k=1}^{K} \frac{p_\Theta(\boldsymbol{x}_{1:M}|\boldsymbol{z}^k)(p(\boldsymbol{z}^k))^\beta}{(q_\Phi(\boldsymbol{z}^k|\boldsymbol{x}_{1:M}))^\beta} \right]$$

## C  TECHNICAL DETAILS FOR DATASETS AND METRICS

### C.1  DATASETS

In this work, we report experimental results on three multimodal datasets introduced in previous works, namely PolyMNIST (Sutter et al., 2021), MNIST-SVHN (Shi et al., 2019) and CUB Image-Captions (Shi et al., 2019; Netzer et al., 2011).

PolyMNIST is a synthetic dataset consisting of five modalities, depicting MNIST digits sharing the digit label, patched on random crops from five different background images, one for each modality. Each datapoint consists of five images, one per modality, where all the images share the digit label, but not the style of the handwriting, as MNIST samples are shuffled prior to the pairing. Shared information between modalities is therefore the digit label, while modality-specific information includes the style of the handwriting and the specific background image.

Similar in nature to PolyMNIST, the MNIST-SVHN dataset consists of paired MNIST (LeCun et al., 2010) and SVHN (Netzer et al., 2011) samples, which share the digit label. As with PolyMNIST, the challenge for the model consists in separating shared digit content from private visual features characterising each modality. We use this dataset only in Appendix F.

Finally, the Caltech Birds (CUB) Image-Captions dataset consists of images of birds paired with matching linguistic descriptions. Making this experiment ambitious is the high amount of modality-specific information for each modality, due to the different nature of the information sources. Moreover, while for PolyMNIST the amount of shared information for each sample is fixed, always consisting of the digit label, here the amount of shared information present can vary from datapoint to datapoint. In fact, descriptions can be more or less detailed, leading to more or less shared information between modalities. This dataset presents a lot of challenges typical of a realistic experiment, which makes it a relevant setting to benchmark existing approaches.

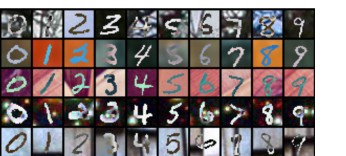 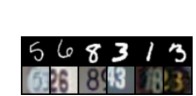 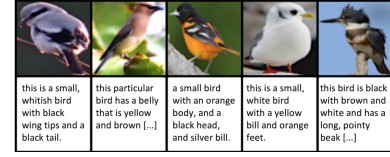

Figure 5: Illustrative samples from PolyMNIST (left), MNIST-SVHN (center) and CUB Image-Captions (right) respectively. Images taken from Shi et al. (2019) and Daunhawer et al. (2022).

### C.2  METRICS

In our experiments, we evaluate models in terms of generative quality and generative coherence. To evaluate generative quality we adopt the FID score (Heusel et al., 2017), a state-of-the-art metric to quantify visual sample quality for generative models in image domains, which has been shown to correlate well with human judgement. To compute FID scores we use the implementation from Seitzer (2020).

To compute generative coherence in the PolyMNIST and the MNIST-SVHN experiments, as done in previous works (Shi et al., 2019; Sutter et al., 2021; Daunhawer et al., 2022), we use the training samples for each single modality to train a classifier for the digit label. The resulting classifiers are then used to obtain a quantitative measure for conditional and unconditional generative coherence. To measure conditional generative coherence from a given input modality $m$ to a given target modality $n$, we feed the conditionally generated sample to the classifier for the target modality $n$, then compare the predicted digit label with the true digit label. The matching rate for the whole test

set measures conditional generative coherence from modality $m$ to modality $n$. To obtain a single representative metric for PolyMNIST, we compute the average coherence for each target modality, and then average the five obtained results. For unconditional generation, coherence is measured by feeding jointly-generated samples for each modality to the corresponding trained classifier. Then the measure for coherence is the rate with which all classifiers output the same predicted digit label for $N$ generations, where $N$ is equal to the length of the test set, for compatibility with the sample size used to compute conditional coherence.

## D  TECHNICAL DETAILS FOR THE EXPERIMENTS

**PolyMNIST**   Following the suggestions by the authors in Shi et al. (2019) to reach best performance on the MNIST-SVHN dataset, which is very similar to the PolyMNIST dataset in nature, we train the MMVAE assuming Laplace priors, likelihoods and posteriors, constraining their scaling across the D dimensions to sum to D. For compatibility, we use the same settings for the MMVAE+. Following recent work (Daunhawer et al., 2022), for MVAE and MoPoE-VAE we assume Laplace likelihoods and Gaussian priors and posteriors, which enable to compute product distributions in closed form. For a fair comparison with MMVAE+, we also use Laplace likelihoods for MMJSD. The size of the latent space for MVAE and MoPoE-VAE on PolyMNIST is set to $512$ dimensions, while we opt for a much smaller space of $64$ dimensions when training both MMVAE and MM-VAE+, partially due to memory constraints coming from the use of multi-sample estimators. For the results in section 4.1.1, we split the latent space capacity equally between shared and private subspaces for MMVAE+. As in Daunhawer et al. (2022), we train both MVAE and MoPoE-VAE for 500 epochs, while MMVAE, MMVAE+, as well as MMJSD, are trained for 50 epochs. Both MMVAE+ and MMVAE are trained with the $K$-sample version of the respective objectives and $K = 10$, using the DReG estimator, as explained in Appendix B. In section 4.1.1, we show results for MVAE, MMVAE, MoPoE-VAE and MMVAE+ for a range of representative values of the regularization hyperparameter $\beta$. For MVAE and MoPoE-VAE we choose $\beta \in \{0.03, 0.3, 1.0, 3.0\}$, while for MVAE and MMVAE+ we instead use $\beta \in \{0.5, 1.0, 2.5, 5.0\}$. Finally, we want to point out that in our ablations on the PolyMNIST experiment we have witnessed sporadic optimization problems for MMVAE and MMVAE+ with some runs diverging. We suspect this behaviour can be due to a relatively small latent space capacity for five modalities, or rather caused by using a too high learning rate. In future work we plan to investigate this issue further, in order to understand the rationale behind it and resolve it.

**MNIST-SVHN**   We introduce this experimental setting only in Appendix F to compare MMVAE+ with another example of existing models with separate shared and private latent representations, namely DMVAE (Lee & Pavlovic, 2021). For a fair comparison, we adopt the same choices for network architectures with MMVAE+ that the authors adopt with DMVAE. In particular both models are trained for 10 epochs with MLP encoders and decoders for the MNIST modality and CNN encoders and decoders for the SVHN modality. For MMVAE+ we adopt the $K$-sample version of the objective with $K = 10$ and the DReG estimator.

**CUB Image-Captions**   On this experiment, for all compared models, we assume Laplace and one-hot categorical likelihood distributions for images and captions respectively, with Gaussian priors and posteriors, and set the total latent space capacity to $64$ dimensions. We train MVAE and MoPoE-VAE for $150$ epochs as done in previous work (Daunhawer et al., 2022), while MMVAE and MMVAE+ are trained for $50$ epochs with the $K$-sample version of the respective objectives and $K = 10$, using the DReG estimator. Following Daunhawer et al. (2022), to achieve the best results, we train MVAE and MoPoE-VAE with $\beta = 9.0$. MMVAE and MMVAE+ instead are trained with $\beta = 1.0$.

**General specifics for the experiments**   All models are trained using the Adam optimizer (Kingma & Ba, 2014). We choose a learning rate of 5e-4 for MVAE and MoPoE-VAE, following Daunhawer et al. (2022), while for MMVAE and MMVAE+ we choose 1e-3. DMVAE and MMJSD are also trained with learning rate 1e-3, as suggested in the respective implementations by the authors. All quantitative results are averaged over multiple seeds, and reported with standard deviations.

# E    ADDITIONAL RESULTS

In this section we report additional or more extensive qualitative and quantitative results for MM-VAE+ and the compared models.

## E.1    POLYMNIST

**Conditional and unconditional generation**    Here we report additional and more extensive qualitative results for the comparison between MMVAE+, MVAE, MMVAE and MoPoE-VAE in section 4.1.1. In particular, Figures 6 and 7 report conditional and unconditional generation results respectively.

**Conditioning on private information**    By learning shared and private information disentangled in separate latent subspaces MMVAE+ offers the possibility of conditioning on modality-specific features for generation, which is not feasible for models with a single latent space and can be important in practice. The showcased results in Figures 8 and 9 are yet another demonstration of the simultaneous modelling of detailed private variation and coherent shared content between modalities.

**Latent traversals**    To explore variation in single dimensions of the latent space, we compute latent traversals following Shi et al. (2019). We perturb the embedding $\mu_m = (\bar{\boldsymbol{u}}_m, \bar{\boldsymbol{w}})$ prior to conditional generation[3], where $\bar{\boldsymbol{u}}_m$ is the mean shared embedding inferred from modality $m$ and $\bar{\boldsymbol{w}}$ is sampled from $p(\boldsymbol{w})$, along each dimension $d$ linearly in the range $(\mu_m^d - 5\sigma, \mu_m^d + 5\sigma)$. $\sigma$ is the standard deviation of the prior distribution on the latent code, which in our case is equal to one. Figure 10 shows the results for latent space traversals. Perturbing $\bar{\boldsymbol{u}}_m$ only affects shared content between modalities, and changes in the represented digit are consistent for all generated modalities. Perturbing $\bar{\boldsymbol{w}}$ results in changes to modality-specific features. Compared to previous work (see qualitative results in Shi et al. (2019)), single dimensions do not exhibit simultaneous variation in private and shared information, and when changes in the content are witnessed, the represented digit changes consistently in all generated modalities. The results show that information in the latent space is well-disentangled across single dimensions, which is a proxy for the quality of representations towards downstream tasks.

**Latent classification accuracy**    An additional comparison of the models can be based on the quality of learnt representations. Previous works (Sutter et al., 2021; Daunhawer et al., 2022; Shi et al., 2019) have contributed to the establishment of latent classification accuracy as the most prominent metric to assess the quality of learnt representations for multimodal VAEs. This metric assesses the amount of shared information that can be retrieved by a linear classifier from the latent embeddings, and it is a proxy for how well shared and private information is disentangled in the latent space. In Figure 11, we compare MMVAE+, MVAE, MMVAE and MoPoE-VAE in terms of latent classification accuracy: in particular we report the mean of the accuracies obtained with inference from each single modality for different degrees of regularization. MMVAE+ achieves the best score among the four models on this metric. In addition we use this metric to prove that the MMVAE+ achieves factorization between shared information in $\boldsymbol{u}$ and private information in $\boldsymbol{w}$ in the latent space. In particular given $\boldsymbol{u}_m \sim q_{\phi_m}(\boldsymbol{u}|\boldsymbol{x}_m), \boldsymbol{w}_m \sim q_{\phi_m}(\boldsymbol{w}|\boldsymbol{x}_m)$ latent representations encoded from a given modality $m$, we compute latent classification accuracies by training digit classifiers on $\boldsymbol{u}_m$, $\boldsymbol{w}_m$ and $(\boldsymbol{u}_m, \boldsymbol{w}_m)$ embeddings respectively. The showcased results in Figure 11 show that all the shared information between modalities present in the latent code is present in the $\boldsymbol{u}$ subspace, which validates our modelling assumptions for MMVAE+. The accuracies obtained with classifiers trained on private embeddings are, as expected, markedly low. These results offer additionl empirical evidence demonstrating that MMVAE+ learns representations that factorize into shared and private subspaces consistently with respect to changes in hyperparameters.

---

[3]Note that only for this analysis, instead of sampling $\boldsymbol{u}_m$ from $q_{\phi_m^u}(\boldsymbol{u}|\boldsymbol{x}_m)$ for conditional generation, we take the mean of the distribution $\bar{\boldsymbol{u}}_m$. This is consistent with the computation of latent traversals in Shi et al. (2019). $\bar{\boldsymbol{w}}$ is instead sampled from $p(\boldsymbol{w})$ and it is not the mean of the distribution, consistently with our computation of conditional generation.

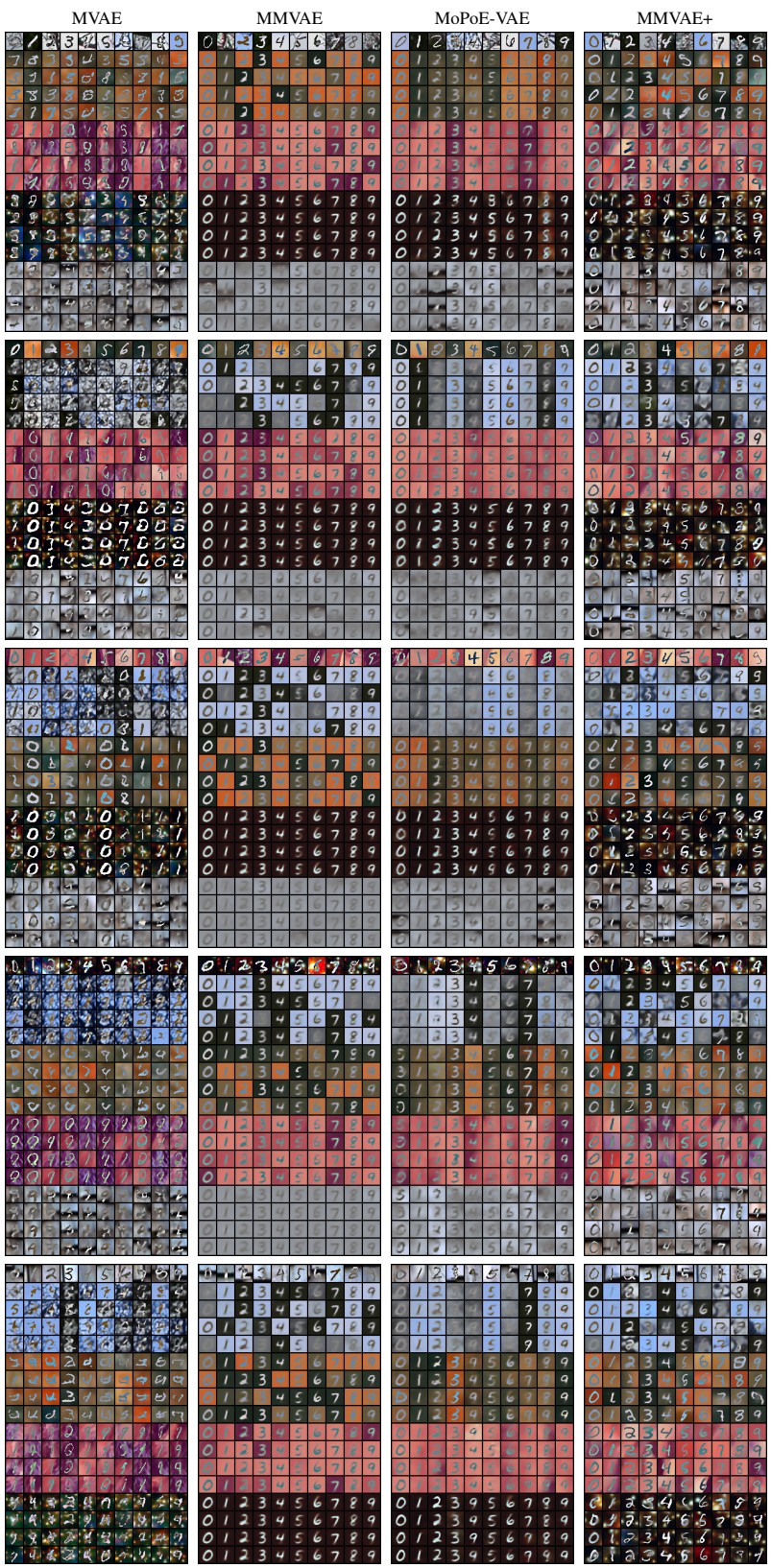

Figure 6: Conditional generation extensive results for MMVAE+ compared with MVAE, MMVAE and MoPoE-VAE. For each different starting modality, we show input samples in the top row, and four instances of conditional generation for the remaining target modalities. $\beta = 1$ for all compared models.

MVAE

MMVAE

MoPoE-VAE

MMVAE+

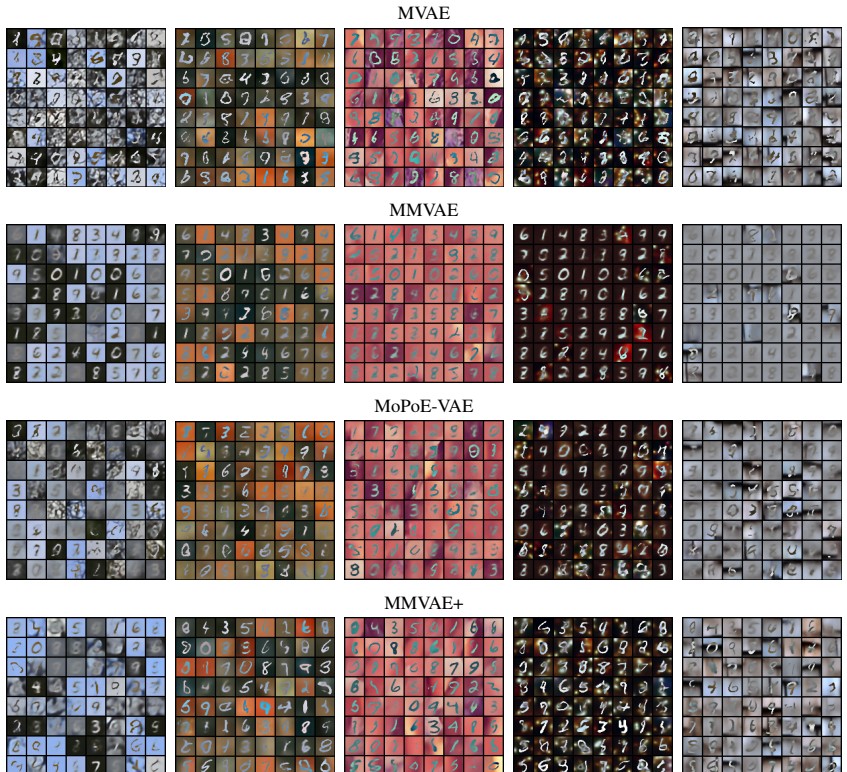

Figure 7: Unconditional generation using prior samples for MMVAE+, MVAE, MMVAE and MoPoE-VAE. For each model we draw $64$ prior samples for the latent code and generate in each of the five modalities. $\beta = 1$ for all compared models.

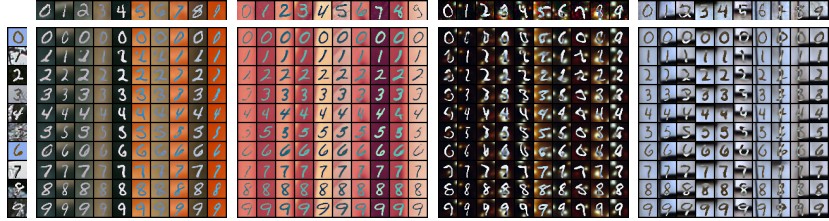

Figure 8: MMVAE+ on PolyMNIST: $m_0 \rightarrow m_{1,2,3,4}$ generation conditioning on shared encoded information from input samples on the left column, and private encoded information from target samples in the top row.



Figure 9: MMVAE+ on PolyMNIST: generation conditioning on private encoded information from the samples in the top row, with shared information common to all modalities sampled from the unconditional prior.

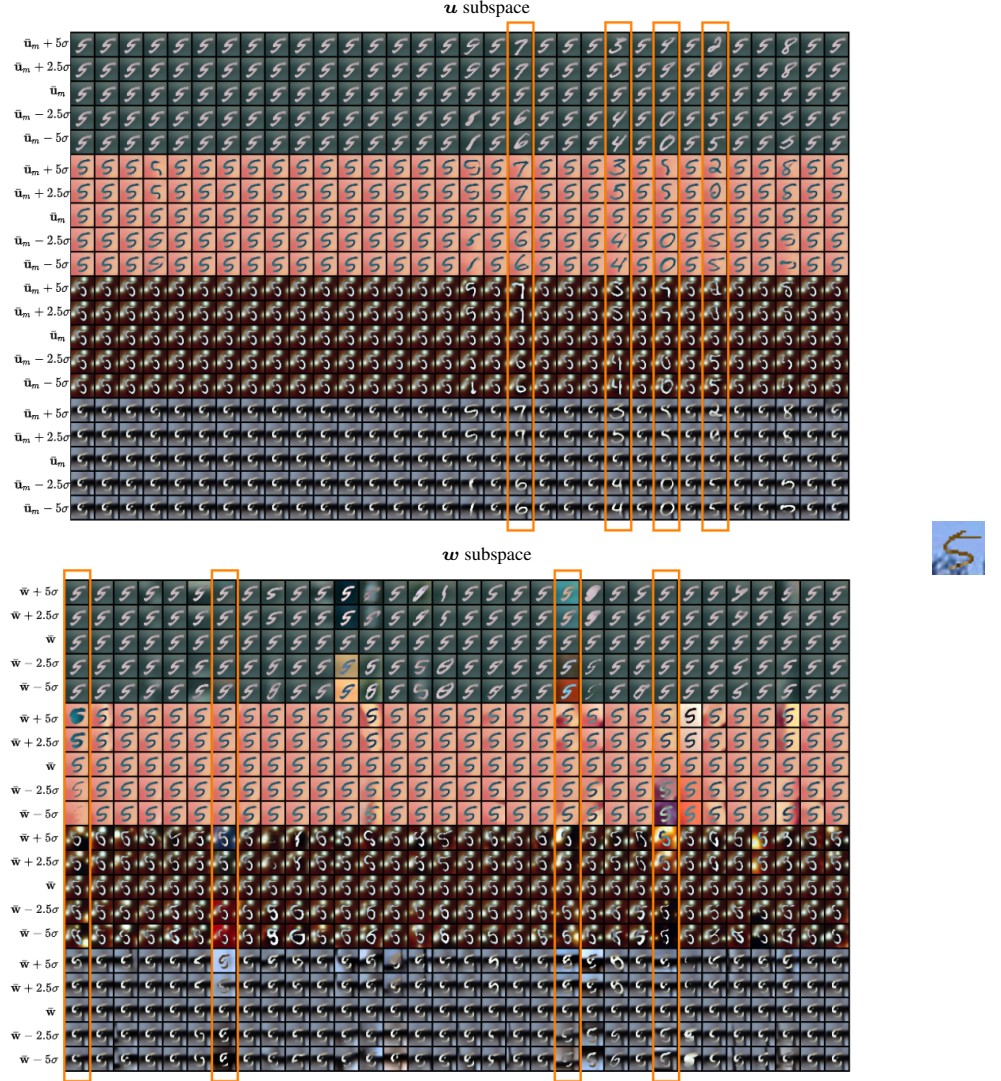

Figure 10: MMVAE+: latent traversals on PolyMNIST. Input sample (right) from $m_0$ and conditional generation results perturbing $\bar{u}_m$ (top) and $\bar{w}$ (bottom) along each single latent dimension. Each column corresponds to a single dimension in the latent space. We highlight some columns that correspond to latent dimensions encoding relevant features, from what this qualitative analysis indicates.

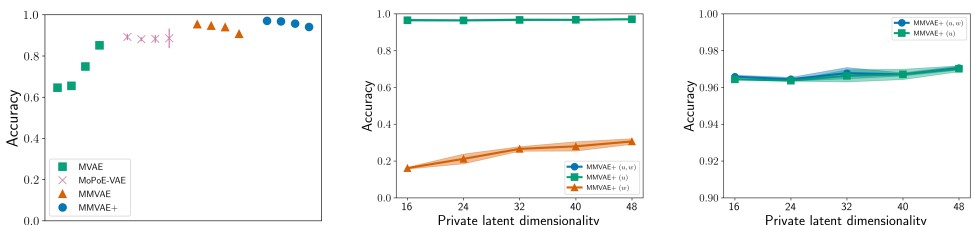

Figure 11: *Left*: Latent classification accuracy results for MMVAE+, MVAE, MMVAE and MoPoE-VAE compared, for a range of representative values of the regularization hyperparameter $\beta$. *Center*: MMVAE+ latent classification accuracies varying private and shared latent dimensionality. $\beta = 1$ and total number of latent dimensions fixed to $64$. *Right*: Zoomed-in version of the plot in the center.

### E.2 CUB IMAGE-CAPTIONS

Here we report additional and more extensive results for the comparison between MMVAE+, MVAE, MMVAE and MoPoE-VAE in section 4.2 on the CUB Image-Captions dataset.

**Image generation** Figure 12 and 13 report unconditional and conditional image generation results for MMVAE+, MVAE, MMVAE and MoPoE-VAE on the CUB Image-Captions dataset. Image generation results for MMVAE and MoPoE-VAE in this complex experiment are characterized by the presence a considerable number of generated samples for which modality-specific features are collapsed to average values, resulting in completely blurred images. Images of this kind are a significant part of the ones produced with unconditional generation, and the totality of the ones produced with conditional generation. MMVAE+ results on the contrary do not exhibit signs of this pattern, which indicates good modelling of modality-specific information even in this complex setting. In addition, conditional generation shows important coherence results for this complex experiment.

**Text generation** For completeness, here we showcase unconditional and conditional text generation results for MVAE, MMVAE and MoPoE-VAE compared with MMVAE+ on the CUB Image-Captions experiment. While matching color with the descriptions indicates some generative coherence for MMVAE+, improving the quality of captions should be an important focus for future work. Note that for MMVAE+ we have used convolutional encoder and decoder networks for the next modality. Switching to Resnets, or other more powerful network architectures for text generation, could improve the quality of produced captions, as we plan to investigate in future work.

**Latent traversals** In the same way as in the PolyMNIST experiment (see E.1) we compute latent traversals for conditional image generation with MMVAE+ on the CUB Image-Captions dataset. This serves to understand how information is encoded in the latent space in a more complex setting.

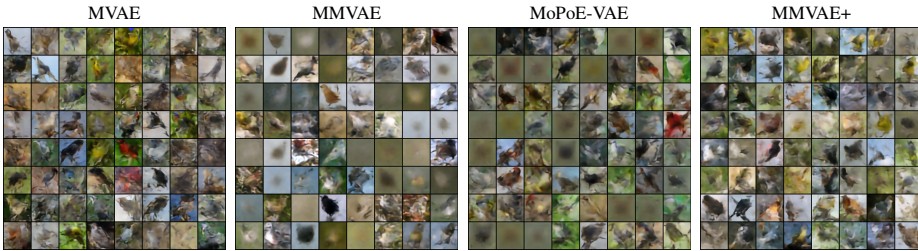

Figure 12: Unconditional image generation on the CUB Image-Captions dataset using prior samples, for MMVAE+, MVAE, MMVAE and MoPoE-VAE. For each model we draw 64 prior samples for the latent code and generate in the image modality.

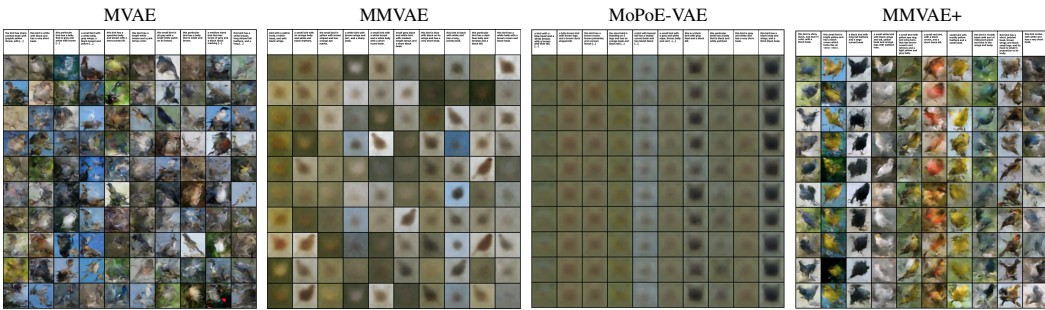

Figure 13: Conditional caption-to-image generation extensive results on the CUB Image-Captions dataset. Input captions in the top row, and ten instances of conditional image generation below.

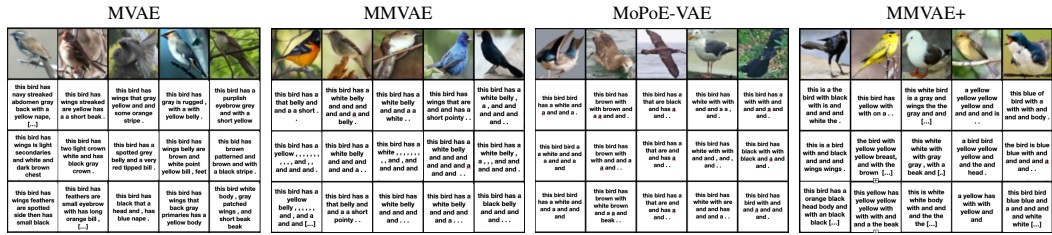

Figure 14: Conditional text generation on the CUB Image-Caption dataset. Input images in the top row, and three instances of conditional text generation below.

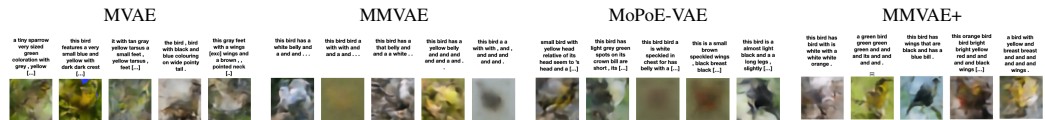

Figure 15: Unconditional text generation on the CUB Image-Caption dataset. Captions generated from five different samples drawn from the unconditional prior on latent representations. Images generated from the same latent code are also shown below the corresponding caption.

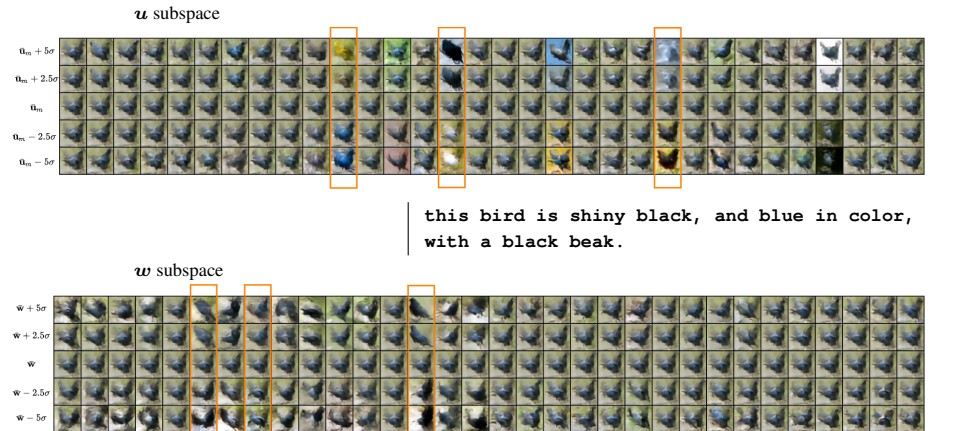

Figure 16: MMVAE+: latent traversals on CUB. Input caption (right) and conditional image generation results perturbing $\bar{u}_m$ (top) and $\bar{w}$ (bottom) along each single latent dimension. Each column corresponds to a single dimension in the latent space. We highlight some columns that correspond to latent dimensions encoding relevant features, from what this qualitative analysis indicates.

## F    COMPARING MODELS WITH SEPARATE SHARED AND MODALITY-SPECIFIC LATENT REPRESENTATIONS

In this section we want to compare MMVAE+ with existing multimodal VAE models which uses separate shared and modality-specific latent subspaces to learn factorized representations. We want to further elaborate on the shortcut problem that existing models of this kind present, from which our model, notably, does not suffer.

In section 4.1.2 we have showed MMJSD (Sutter et al., 2020) performance on PolyMNIST compared with MMVAE+, showing how the former needs strict regularization to achieve coherence in shared content between modalities. In this Appendix we show that this limitation is common to other existing models with separate shared and private latent representations. As an example we

take the DMVAE (Lee & Pavlovic, 2021), which extends the MVAE objective to include separate modality-specific latent subspaces in addition to a shared latent subspace. We report an ablation for this model on the MNIST-SVHN experiment Shi et al. (2019), which appeared in the original work, in which we test the effect of varying the number of latent dimensions allocated to private subspaces for each modality.

In particular, in the original work the number of dimensions for the shared latent space is set to 10, while the number of dimensions for the private subspace for MNIST and SVHN modality are respectively set to 1 and 4. To test the stability of the model with respect to changes of latent dimensionalities we augmented gradually the number of dimensions for the SVHN private latent subspace to 6, 8, 10 and 12, keeping the number of dimensions for the shared latent space fixed to 10. The number of dimensions for the MNIST private latent subspace were also changed accordingly, to 3, 5, 7 and 9 respectively. The results in the left plot of Figure 17 show that, as soon as more capacity is given to modality-specific subspaces, coherence abruptly decreases and soon reaches values obtained for random guess. This shows that the DMVAE presents the same problem uncovered for MMJSD. Both these models need to have constraints on private latent subspaces, either in the form of reduced number of dimensions or regularization, to achieve generative coherence. In other words, given enough capacity for modality-specific latent subspaces and without strong regularisation, all the information ends up being encoded independently in such subspaces, and the shared subspace is ignored by decoders.

MMVAE+ on the other hand does not suffer from this limitation. As ablations on PolyMNIST and MNIST-SVHN demonstrate, shared information across modalities is modelled in the latent space consistently with respect to changes in hyperparameters controlling latent dimensionality and regularization. This is a key difference of MMVAE+ from previous works which is crucial in practice, especially when dealing with complex experimental settings.

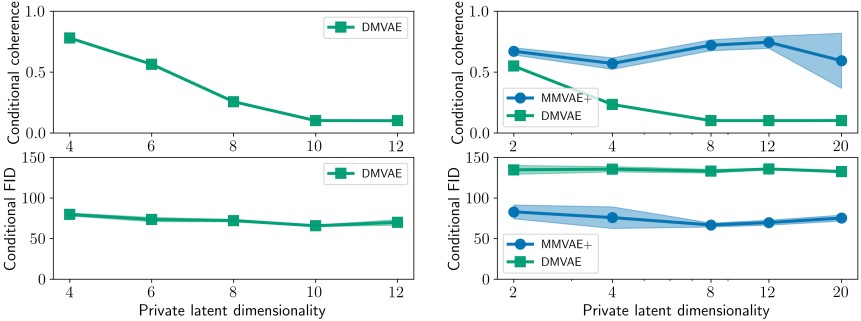

Figure 17: MMVAE+ compared with DMVAE on the MNIST-SVHN experiment. Left: DMVAE conditional coherence and FID for the SVHN modality, progressively augmenting private latent dimensionality from the values used in the original work. Right: MMVAE+ and DMVAE compared, with the same value for the regularization hyperparameter $\beta = 2.5$, varying private latent dimensionality in the same way for both models.

# G PRIOR PRIVATE INFORMATION FOR CROSS-MODAL RECONSTRUCTION AND CONDITIONAL GENERATION

In this section we want to elaborate on resampling strategies for the private latent code for cross-modal reconstruction at training time and conditional generation and test time. At training time we resample private information for cross-modal reconstruction from auxiliary priors, for which we parametrize the variance. As private features are not predictable across modalites, in practice these priors tend to be very sharp and centered around average values, as confirmed by the results obtained using these priors for resampling private information for conditional generation at test time in Figure 18. Instead, to obtain high-quality modality-specific information at test time, for conditional generation we resample private information $\boldsymbol{w} \sim p(\boldsymbol{w})$. Interestingly, at test time, one can parameterize the scale of the $p_{\alpha}(\boldsymbol{w})$ distribution by a multiplying factor $\alpha$ when resampling private information. A sharper prior results in average-looking generated samples, while with a

more flat prior generative quality is improved. The accurate separation between shared and private information in the latent space, allows to tune visual quality of generated samples maintaining a high-level of semantic coherence. This allows to obtain the best generative quality, satisfying the level of coherence that is needed for a given application, by fine-tuning $\alpha$ at test time, while for other models a similar procedure would require training several instances with different hyperparameter values.

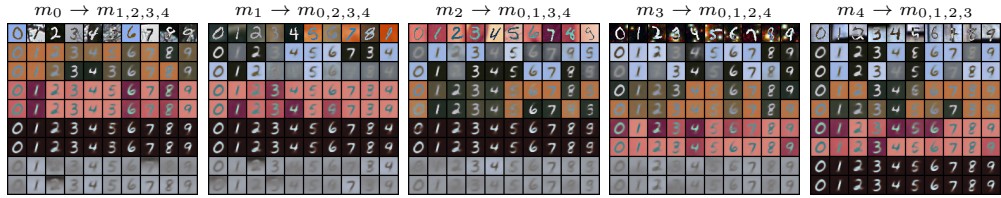

Figure 18: MMVAE+ conditional generation results on PolyMNIST obtained by resampling private information from $p_{\psi_n}(\tilde{\boldsymbol{w}}_n)$ for each given target modality $n$.

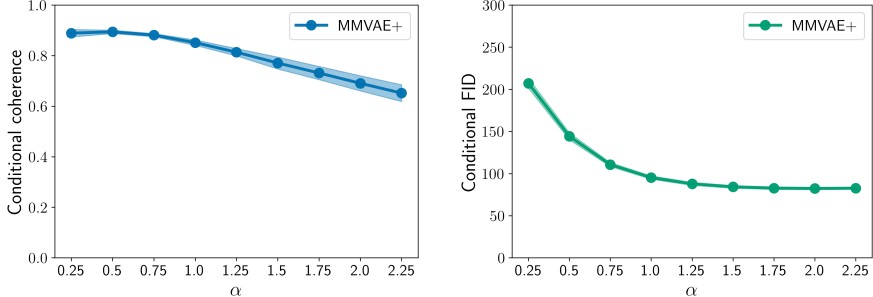

Figure 19: MMVAE+: Conditional generative coherence (left) and FID score (right) on PolyMNIST, varying $\alpha$, multiplicative factor for the scale parameter of the $p_\alpha(\boldsymbol{w})$ distribution.

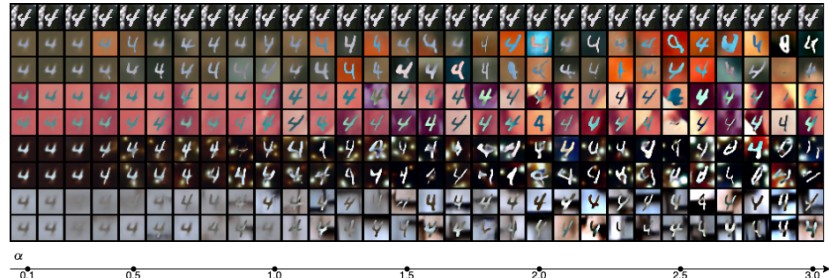

Figure 20: MMVAE+ on PolyMNIST: conditional generation $m_0 \to m_{1,2,3,4}$ for a single sample, varying $\alpha$, multiplicative factor for the scale parameter of the $p_\alpha(\boldsymbol{w})$ distribution. Each column corresponds to a given $\alpha$ value. $\alpha$ values shown for the interval $[0.1, 3.0]$ with step $0.1$, increasing left to right. A single test input sample (top row) and two instances of conditional generation for each target modality are shown.

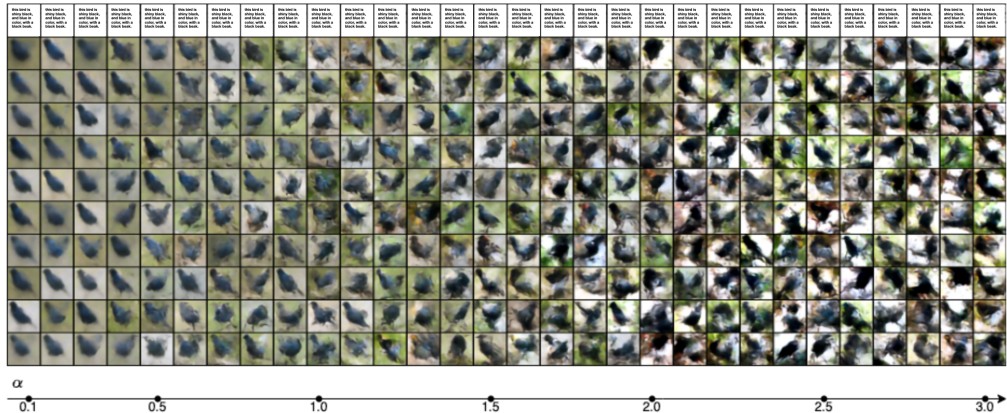

Figure 21: MMVAE+ on CUB Image-Captions: conditional caption-to-image generation for a single sample, varying $\alpha$, multiplicative factor for the scale parameter of the $p_\alpha(\boldsymbol{w})$ distribution. Each column corresponds to a given $\alpha$ value. $\alpha$ values shown for the interval $[0.1, 3.0]$ with step $0.1$, increasing left to right. A single test input caption (top row) and ten instances of conditional image generation are shown.

