# OpenReview forum: "MMVAE+: Enhancing the Generative Quality of Multimodal VAEs without Compromises"
_ICLR.cc/2022/Workshop/DGM4HSD — ICLR 2022 DGM4HSD workshop Poster_

### Official Review · Reviewer_FECh · 2022-03-15
**I have no major concerns and recommend acceptance of this paper**

**Rating:** 8
**Confidence:** 3

**Review:**

This paper proposes a novel method MMVAE+ for weakly-supervised generative learning with multiple modalities. It is a variant of the mixture-of-experts multimodal VAE (MMVAE, Shi et al., 2019). The main goal of this method is to enhance the generative quality without a non-trivial compromise between generative quality and generative coherence. It is achieved by learning shared information by optimising cross-modal reconstruction does not conflict with modeling private information. Experiments on PolyMNIST, Caltech Birds and Image-Captions show the superior of this method.

Overall, this paper is of high quality. It is written very well and technically sound. The theoretical analysis and experiments are very comprehensive. I have no major concerns and recommend acceptance of this paper.

---

### Official Review · Reviewer_444W · 2022-03-18
**Improved method for multi-modal VAEs**

**Rating:** 6
**Confidence:** 3

**Review:**

This paper addresses the challenge of modeling data from multiple modalities or datasets. This can be done within the framework of variational autoencoders by introducing latent variables specific to each modality. This changes the generative model p(x|z) to p(x|z, u) for the new modality specific variables, u. The paper describes a variety of prior methods which augment VAEs in this way to model multi-modal data. The authors improve on this prior work by introducing private latent variables (information specific to each modality) and public latent variables (information shared across modalities) which they claim leads to improved sample quality.

Overall, I found this paper difficult to understand. I am not familiar with the prior methods in the space and I do not feel that they were adequately described. As well I found the method somewhat confusing. Equation 1 is pretty involved and I don't think the text explains it very well. I had to go through the appendix to begin to understand it and this should not be necessary. Experimental details can be in the appendix, but information key to understanding what's new about this method should be in the main body.

It appears that the proposed method leads to improvements in image quality compared to prior multi-modal generative models. While these results appear valid, likelihoods (or elbos) should also be compared as they can easily be computed with all models proposed here. This would give a reader who is not as well versed in this area a more understandable evaluation.

While I had difficulty understanding the method, I would advocate for its acceptance if the authors expanded on the differences between their method and prior work and explained the method in a bit more technical detail.

---

### Official Review · Reviewer_b1e6 · 2022-03-25

**Rating:** 6
**Confidence:** 4

**Review:**

The paper studies an improvement of mixture-of-experts multimodal VAE.
The main contribution of the training objective is revising the $\beta$ regularizer for the MMVAE training. The authors also provide some theoretical justification.

Overall, the results are good, but some other non-VAE based generative modeling baselines may consider incorporated in the future version as additional baseline (e.g., GAN-based and seq2seq modeling)

---

### Decision · Program_Chairs · 2022-03-26

Accept (Poster)